# Remote Implementation of a School-Based Health Promotion and Health Coaching Program in Low-Income Urban and Rural Sites: Program Impact during the COVID-19 Pandemic

**DOI:** 10.3390/ijerph20021044

**Published:** 2023-01-06

**Authors:** Liana Gefter, Nancy Morioka-Douglas, Ashini Srivastava, Can Angela Jiang, Sonal J. Patil, Eunice Rodriguez

**Affiliations:** 1Division of Primary Care and Population Health, Department of Medicine, Stanford University School of Medicine, Palo Alto, CA 94304, USA; 2Cleveland Clinic Community Care, Cleveland Clinic, Cleveland, OH 44195, USA; 3Division of General Pediatrics, Department of Pediatrics, Stanford University School of Medicine, Palo Alto, CA 94304, USA

**Keywords:** vulnerable youth, child health, adolescent health, chronic diseases, community health, health communication, health educators, nutrition and diet, school health instruction, remote implementation, school-based health promotion, rural and urban health prevention programs

## Abstract

Background: Adapting existing health programs for synchronous remote implementation has the potential to support vulnerable youth during the COVID 19 pandemic and beyond. Methods: The Stanford Youth Diabetes Coaches Program (SYDCP), a school-based health promotion and coaching skills program, was adapted for remote implementation and offered to adolescents from low-income communities in the US: an urban site in San Jose, CA and rural sites in Lawrence County, MO, and Central Valley, CA. Participants completed online pre- and post- surveys. Analysis included paired T-tests, linear regression, and qualitative coding. Results: Of 156 enrolled students, 100 completed pre- and post-surveys. Of those: 84% female; 40% Hispanic; 37% White; 28% Asian; 3% African American; 30% other race. With T-tests and regression models, the following measures showed statistically significant improvements after program participation: health knowledge, patient activation, health understanding and communication, consumption of fruits and vegetables, psychosocial assets of self-esteem, self-efficacy, problem-solving, and ability to reduce stress. Technology barriers were frequently reported at Lawrence County site. 96% participants reported making a lifestyle change after program participation. Conclusions: Remote implementation of health promotion programs for vulnerable youth in diverse settings has potential to support adoption of healthy behaviors, enhance patient activation levels, and improve psychosocial assets.

## 1. Introduction

The burden of chronic disease across the globe is rising daily and disproportionately affects individuals from low-income and ethnic minority populations [1,2]. The impact of this burden extends to youth as seen by the 4.8% relative annual increase in the incidence of type 2 diabetes in youth (ages 10–19) in the US [3]. Thus, providing opportunities for ethnic minority youth and youth from low-income communities to become engaged with healthcare and empowered to improve their own health is particularly important.

The COVID-19 global pandemic highlighted numerous ways in which inequality leaves marginalized groups more vulnerable to disease and poor health outcomes [4]. Specifically, in the US, African American and Latino individuals in the early stages of the pandemic experienced a disproportionate burden of COVID-19 cases and COVID-19 related poor health outcomes, including mortality [5]. As a result of the pandemic, public health experts and health educators have had to rethink mechanisms for engaging underserved populations in health promotion [6]. In addition to the acute need for health promotion programs related to COVID-19, there also exists a pressing need to address the chronic health conditions that are major risk factors for poor outcomes from COVID-19 [7]. Programs that take action to support underserved populations to manage and prevent chronic disease are an investment in post-pandemic health and the health of future generations [8].

For youth, the pandemic has resulted in widespread challenges to physical and mental health related to isolation stress, childcare deficits, reduced access to nutritious food, decreased opportunities for physical activity, and decreased insurance coverage [9]. For example, the increased risk of childhood obesity during the pandemic has been well documented [10,11]. For youth in low-income and/or racial and ethnic minority populations, these challenges are all compounded because, together with their families, they have faced higher risk of viral contagion, are more exposed to deaths, and are more vulnerable to the social and economic consequences of the pandemic [12].

To best support vulnerable youth during this time, experts recommend facilitating avenues of communication and empowerment through school and community-based programs [13]. When in-person programming is not an option due to pandemic restrictions, geographic barriers, transportation challenges, or other obstacles, synchronous remote program implementation has potential to support youth who otherwise would not have access to such opportunities [14]. In fact, it is widely understood that e-Health (defined by the World Health Organization as the “transfer of health resources by electronic means” [15]) is an important tool to connect medically underserved populations with health information [16]. Evidence suggests adolescents respond well to online health education courses [17] and that these programs can have positive impacts on children’s health [18,19,20]. Yet, researchers are just beginning to learn about the impact of providing remote health promotion programs [21], particularly in economically disadvantaged communities.

In this study, the authors assess the implementation of a school based, health promotion program adapted for synchronous remote learning for adolescent participants from rural and urban underserved communities. The validated health promotion program, called the Stanford Youth Diabetes Coaches Program (SYDCP), has been implemented widely over the past twelve years with community partners as an in-person program in 12 US states and in Canada [22,23,24]. In a previous study of program efficacy, using a quasi-experimental controlled design, the authors demonstrated that this particular intervention produces significant health related benefits for vulnerable youth when provided in-person [22,23].

Although the SYDCP has been shown to provide significant benefit to participants when implemented in person, in this study the authors aim to assess whether remote participation can also benefit and support youth to improve and promote key health related outcomes as described in the methods section. This study aims to understand whether remote participation is associated with benefit for participants by comparing responses between pre and post participation surveys.

## 2. Materials and Methods

### 2.1. Program Description

The Stanford Youth Diabetes Coaches Program (SYDCP) is a validated “train the trainer program” in which health care professionals and trainees teach healthy high school students (grades 9–12; ages 14–18) from underserved schools to coach family members with chronic health conditions [22]. The curriculum is based on Kate Lorig’s Adult Chronic Disease Self-Management Model [25], Social Cognitive Theory [26], and peer health coaching [27], and is designed to address the burden of chronic disease in underserved communities by focusing on health knowledge, communication skills, goal setting, problem solving, and healthy behaviors. The program curriculum consists of eight highly structured and interactive one-hour lessons that are taught once a week for eight weeks and include health knowledge, as well as training in coaching skills and problem-solving. Participants complete a coaching assignment with the family member they are coaching each week. Instructors access orientation materials online.

### 2.2. SYDCP Adaptation for Remote Implementation

In March 2020, the SYDCP team rapidly adapted the SYDCP curriculum from an in-person program into a completely remote program that delivered synchronous virtual lessons through an online portal. A SYDCP research team member observed each of the pilot remote classes and took notes on student participation to adjust and improve the SYDCP curriculum accordingly. By September 2020, the curriculum was revised to include features designed to engage students remotely including: setting ground rules for remote participation etiquette; reducing text content of program slides; utilizing the chat feature to have participants respond to quizzes and discussion questions; enabling role plays by asking volunteers to unmute; asking participants to use objects at home to learn subjects like reading nutrition labels; and encouraging instructors to verbally address chat comments and questions in real time.

### 2.3. Participant Recruitment

In this study participants were recruited from seven underserved high schools in three geographic sites: San Jose, CA (one high school, urban setting); Central Valley, CA (five high schools, rural setting); and Lawrence County, MO (one high school, rural setting). Recruitment method varied by site, but the goal was to recruit approximately 30 high school students per implementation group. In San Jose, participation in the SYDCP was offered to all high school students (grades 9–12) through widespread program advertisement facilitated by the high schools. All interested high school participants were invited to join the program. In Lawrence County, MO, SYDCP was part of a school class and participation in SYDCP was mandatory for all students enrolled in that class. In the Central Valley, high school teachers at each high school selected and invited high school students whom they believed would benefit most from participation. The sample selected was a convenience sample, based on participant availability and interest in participating in the SYDCP. Researchers received informed consent or assent from all participants.

### 2.4. Program Implementation

The program was offered at each site during the pandemic from September 2020 through June 2021 (Figure 1). One site is considered urban (San Jose, CA), and the other two sites are considered rural (Lawrence County, MO; Central Valley, CA). For purposes of this manuscript, we define “urban” as an area of high population density and infrastructure associated with cities and towns and “rural” as an area of low population density that is located outside of cities and towns [28]. All participants received a program Zoom link and reminder of program meeting time by email. Instructors implemented the program by sharing their screens to deliver program content via slides to the adolescent participants who logged into the class through devices at home. Weekly e-mail reminders were sent to students and residents with the Zoom log-in link and time. The SYDCP team adapted coaching assignments (previously distributed weekly as hard copies during in-person implementation) to Google forms so students could seamlessly complete and submit each week’s coaching assignment. As described below, each participating site differed slightly in the way the program was implemented.

#### 2.4.1. San Jose, CA

The San Jose, CA site remote implementation was part of a long-established partnership between a family medicine residency program and one local underserved high school. The program switched to remote implementation because of the pandemic. Data was gathered from two remote cohorts (Fall 2020 and Spring 2021) that met weekly as an elective after-school program. High school student participants were recruited by a teacher who directs the high school’s medical magnet program, and students were offered hours toward their medical magnet certificate for participation in the program. Family medicine residents taught the sessions as part of their community medicine rotation. The family medicine residents received an introductory email that included logistics for the course, the PowerPoint files for all the classes which include instructor guides, and a short informational video on screen sharing and sound management to support their remote implementation of the program. Attendance and homework were tracked in a spreadsheet to see which students could receive credit for the course, and students accessed the coaching assignments through Google drive.

#### 2.4.2. Central Valley, CA

In Central Valley, CA, SYDCP was taught remotely over Zoom as part of a new partnership through the Central Valley Area Health Education Center (CVAHEC). Data was gathered from three remote cohorts (Spring 2021) that met weekly as an elective after-school program for which the high school students would receive a certificate of completion. The CVAHEC, through their established network of health care provider training programs and high school health educators, was able to recruit family medicine residents and medical students to teach high school students. The CVAHEC also utilized their established relationship with the local school district to identify high school partners that meet our target population criteria. For this pilot, CVAHEC staff distributed recruitment materials to the high schools and coordinated the high school students’ enrollment in the program. The residents and medical students were asked to view the instructional video developed for the San Jose cohort. CVAHEC staff hosted the Zoom class sessions; sent weekly email reminders about the classes; and distributed and collected the coaching assignments on Google drive. A member of the SYDCP team observed each class session to assess fidelity to the model.

#### 2.4.3. Lawrence County, MO

In Lawrence County, MO, the SYDCP was implemented with support from the Missouri Department of Health and Senior Services. Data was gathered from two remote cohorts (Fall 2020 and Spring 2021) that met weekly during school hours as a mandatory activity that counted towards the course grade. Remote implementation of SYDCP was initially planned for at least three rural high schools in Lawrence and Barry Counties, but due to limited availability of health educators and insufficient time to adjust yearly curriculum at other high schools, SYDCP was implemented in one rural high school. The program was taught remotely over Zoom by community health educators. The Fall 2020 session was taught by the local high school health educator. The Spring 2021 session was taught by a health educator from the University of Missouri extension department that works with the local communities. Health educators lived locally and had graduate level professional degrees in health education and public health. Health educators received orientation about the SYDCP curriculum in a one-hour presentation that included information on the curriculum and coaching assignments. During orientation, a nursing doctorate student who had implemented the SYDCP at a rural setting in 2017 shared her experiences and tips for student engagement as well. Health educators routinely communicated with site supervisor (author S.P.) with any questions.

### 2.5. Target Population

The SYDCP aims to reach youth and their families in low-income and/or under-represented and underserved communities. Because of concerns about confidentiality for the youth participants, the research team did not ask youth participants to report on their socio-economic status. Instead, demographics of high schools were researched, and a general understanding of the socio-economic status of youth participants was determined by examining the percentage of students receiving free or reduced lunch at that high school as reported in 2020–2021 [29]. In Central Valley, CA, participants were recruited from five high schools whose average free and reduced lunch percentage is 75%. In Lawrence County, participants were from one high school whose average free and reduced lunch percentage is 50%. In the San Jose, participants were recruited from one high school with a free and reduced lunch percentage of 56%. In Central Valley, CA, school districts, 72% students identified as Hispanic or Latino [30]; in the Lawrence County school district in MO, 38% students identified as Hispanic or Latino [31,32]; and in the San Jose school district, 51% students identified as Hispanic or Latino and 33% as Asian [30].

### 2.6. Outcome Measures

The following outcome measures were assessed: health knowledge; patient activation; health communication and understanding; health behaviors (physical activity, nutrition, stress management); psycho-social assets (self-esteem, self-efficacy, problem solving); and lifestyle changes that promote health. Outcomes and measures used to evaluate these outcomes are highlighted in Table 1.

Participants were asked to complete online pre-surveys before the first program session and online post-surveys immediately after program completion. In addition to basic demographics, the online pre and post surveys included questions to assess the six major outcome measures as described in Table 1.

Health knowledge was assessed with eight previously validated questions from the University of Michigan’s Diabetes Research and Training Center’s Brief Diabetes Knowledge Test [33] and the Spoken Knowledge in Low Literacy in Diabetes scale [34], as well as knowledge questions developed by the SYDCP team derived directly from program curriculum [22].

Patient activation levels were measured with the validated Patient Activation Measure (PAM^®^10) licensed through lnsignia Health 2020 [35]. The PAM^®^10 consists of ten questions that assess knowledge, skills and confidence for self-management of health and healthcare using a Likert scale. Individuals with higher scores are at a higher level of patient activation and demonstrate better health outcomes and healthcare experiences with stronger self-management skills, greater ability to manage stress and higher likelihood to maintain healthy behaviors [40]. PAM^®^10 analysis allots individuals into one of four activation levels along an empirically derived 100-point scale. Individuals in the lowest activation level do not yet understand the importance of their role in managing their own health and have significant knowledge gaps and limited self-management skills. Individuals in the highest activation level are proactive with their health, have developed strong self- management skills, and are resilient in times of stress or change [40,41].

Health communication and understanding was measured with three questions developed by the SYDCP team to gauge participant’s understanding of how to improve their health, communication about health at home, and motivation to adopt healthy behaviors [36].

Health behaviors were assessed with seven questions from the validated California Healthy Kids Survey Physical Health Module 2021 [37] and Stanford Mind and Body Lab [38] that measured self-reported frequency of exercise, consumption of sugary drinks and foods, consumption of high-fat foods, consumption of fruits and vegetables, and hours of sleep.

Psychosocial assets were measured with the validated 10-item Rosenberg self-esteem scale [39] and seven questions from the validated California Healthy Kids Survey 2021 [37] which assess self-efficacy and problem-solving. Lifestyle changes as a result of participation were assessed in the post-survey with open-ended questions developed by the SYDCP team.

Lifestyle change was measured in the post-survey with a direct question about whether the participant made any lifestyle changes during program participation; and an open-ended question asked participants to describe the changes. Post-surveys also included questions developed by the researchers about the person coached and program experience. Multiple-choice questions about ability and ease of connecting to Zoom for online coaching sessions were included to assess barriers to remote implementation at each site.

To better understand our participants’ life experience, pre-surveys also included questions about access to health care services from the validated California Healthy Kids Survey 2021 [37], access to food and transportation from the Carolina Farm Survey, 2013 [42] and whether migrant work was main source of family income from the validated PRAPARE survey, 2016 [43].

### 2.7. Sample Size Determination

Sample size estimations were conducted using results from previous groups to which we provided the program a few years ago and were based on power calculations for specific outcomes where the alpha was set at 0.05 and power at 0.8 based on pre and post-test survey data from previous cohorts. The aim was to observe an effect size of 0.25 SD unit change. Based on these calculations, the estimate was 10 participants needed to see significant change in health knowledge; 195 to see significant change in self-esteem; 101 participants to see significant change in self-efficacy; 162 participants to observe significant change in consumption of fruits and vegetables; 58 participants to see significant change in problem solving and more than 300 participants to see significant changes in physical activity. Sample sizes were estimated using SPSS version 27.

### 2.8. Data Analysis

The authors used paired T-tests and Wilcoxon signed rank tests to compare differences in patient activation, health knowledge, health behaviors, health understanding and communication, and youth assets before and after program participation. To analyze patient activation, the authors tabulated each participant’s score ranging from 0–100 and assigned the participant into one of four levels of activation using an algorithm provided by the PAM^®^ survey developers. The authors first looked at the sample as a whole and then stratified the sample based on location and assessed site differences.

The authors created multivariate linear regression models with change in outcome, calculated as difference in post and pre-test scores, as the dependent variable. The purpose of creating the linear regression models was to assess whether there were differences in outcomes by location when controlling for baseline or pre-test score, grade, and gender. Other independent variables were also explored, but they did not add to the model. Data analysis was conducted using SPSS version 27. Statistical significance was set at *p* = 0.05.

Open-ended survey questions were analyzed using qualitative techniques. Qualitative analysis was done using open and axial coding [44]. Two researchers (AS and LG) read the responses to open-ended questions and independently coded the responses into themes created based on repetition of responses. To minimize bias, an additional double-blind peer review was conducted by another researcher (ER) who independently verified the initial coding categories. In cases of discrepancies in coding, the 3 investigators discussed and reached consensus.

## 3. Results

A total of 156 high school student participants were recruited and completed pre-surveys before starting the program at the three sites. Of these 156, 100 participants had matching and complete post-surveys and were included in the final analysis. Of the 100 participants in the final sample, 34 were from San Jose, CA (response rate 90%), 34 from Central Valley, CA (response rate 69%), and 32 from Lawrence County, MO (response rate 46%) (Figure 1). 84% were female, 70% were 11th and 12th grade students, mean age was 16 years and 40% were Hispanic or Latino (Table 2).

When examined by location, demographic and experiential differences between sites were noted as detailed in Table 2. Most notably, the racial and ethnic identity of participants varied widely between sites. In San Jose, 65% identified as Asian; in Lawrence County, 63% as White; and in Central Valley, 56% as Hispanic or Latino. Participants in San Jose (mean age 15.5 years) were slightly younger than those in Lawrence County and Central Valley (mean age 16.4 years) and were slightly less likely to coach a family member (68%) compared to participants in Lawrence County (78%) and Central Valley (82%). 21% participants in San Jose reported they coached someone who had been diagnosed with diabetes, compared with 31% in Lawrence County and 27% in Central Valley. When examining social determinants of health in all sites, the authors found the majority (77%) of participants at all three sites had easy access and transportation to reach grocery store to shop for food. 93% participants at all three sites had access to fresh fruits and vegetables at most times. While migrant work was not the main source of family income for most participants, in Central Valley 18% of participants reported migrant farm work was the main source of family income (Table 2).

### 3.1. Remote Participation

After participation, 86% participants reported being able to connect to Zoom from home; 78% connected with a computer; 7% with a smart phone; and 2% with an iPad. 25% participants reported experiencing problems when trying to connect to the Zoom classes. Results varied by site with 100% participants in San Jose reporting being able to connect to Zoom from home compared to 91% in Central Valley and 66% in Lawrence County. Accordingly, 91% participants in San Jose reported attending 7 or more of the 8 program sessions compared to 82% in Central Valley and 50% in Lawrence County.

### 3.2. Participants Lost to Follow-Up

Of the 56 participants that either did not complete post-surveys or had incomplete post-surveys, 4 (7%) were from San Jose, 37 (66%) from Lawrence County, and 15 (27%) from Central Valley. When compared to participants that completed both pre and post surveys using Chi square tests, those lost to follow-up were significantly more likely to be male. Additionally, those lost to follow-up were more likely to report migrant work as being the main source of family income and have lower access to food and transportation (See Appendix A in Appendix A). One reason for missing post-surveys was that the program finished at the very end of the school year and links to the post-survey in Lawrence County were sent to participants after the school closed for summer break. A sub-cohort of participants (approximately 50% of the initial cohort) in Lawrence County was lost to follow up because during Fall 2020 there was a surge of COVID cases in the region and many participants in rural areas with poor internet access could not connect to remote classes at home and dropped out of the program. A small number of participants in the Central Valley cohort had to discontinue the program due to scheduling conflicts with sports activities or job commitments.

### 3.3. Outcome Measures

As described in Table 3, when comparing pre and post-survey responses for all participants using paired T-tests, the authors found statistically significant improvements in health knowledge; patient activation; health communication and understanding; health behaviors (consumption of fruits and vegetables, stress management); and youth psycho-social assets (self-esteem, self-efficacy, problem solving). The authors had similar results when pre and post responses were compared using Wilcoxon signed rank test; those results are not reported here.

#### 3.3.1. Health Knowledge

Comparison of composite pre- and post-health knowledge scores demonstrated significant improvements in health knowledge (*p* < 0.01).

#### 3.3.2. Patient Activation

Analysis of difference in pre- and post- test patient activation scores was conducted to assess change in activation levels after program participation. A total of 92 participants completed PAM^®^10 questionnaire satisfactorily and were assigned valid scores and levels based on the algorithm provided by developers. Eight students were excluded from the analysis as they did not receive a valid score due to incomplete responses in either pre-or post-test surveys. Mean pre-test score was 61.49 and mean post-test score was 73.16; the mean difference of 11.66 was highly significant using paired *t*-tests (*p* < 0.001). The PAM^®^ 10 score difference ranged from −21.3 to 50.0 with a median of 9.65. When distributed by levels, participants at baseline in Level 1 (*n* = 5) made the greatest gains and their average PAM^®^ score increased from 43.04 to 67.40 (*p* < 0.013). The PAM^®^ score for participants in Level 2 at baseline (*n* = 18) increased from 50.55 to 62.93 (*p* < 0.004). The average PAM^®^ score for participants in Level 3 (*n* = 56) at baseline increased from 61.94 to 73.96 (*p* < 0.001). For those in Level 4 at baseline (*n* = 13), the average PAM^®^ score increased the least, from 81.85 to 86.11 (*p* = 0.311) as seen in Figure 2.

The authors also examined change in patient activation by levels of activation and cross tabulation of pre and post-test PAM^®^10 levels was significant using Chi square tests (*p* = 0.001).

As seen in Appendix A in Appendix A, 100% of participants who started at Level 1 increased their activation level; 67% participants who started at Level 2 increased their activation level; 43% who started at Level 3 increased their activation level, and 77% participants who started at Level 4 maintained this highest level of activation after program participation.

#### 3.3.3. Health Communication and Understanding

When comparing responses to two questions about health communication and understanding, significant improvements were seen in responses for both questions: communication with family members about health (*p*= 0.012); and understanding of how to improve one’s health (*p* = 0.002).

#### 3.3.4. Health Behaviors

Analysis of changes in health behaviors revealed significant improvements in only two of the areas being assessed. Consumption of fruits and vegetables increased significantly (*p* = 0.004) as did ability to reduce stress (*p* = 0.003). While significant improvements were not found overall in physical activity, consumption of sugary foods and drinks, and consumption of fatty foods, when assessed by site, some significant improvements were seen in these areas.

#### 3.3.5. Psycho-Social Assets

The three youth assets assessed were self-esteem, self-efficacy, and problem solving. Significant improvements were seen across all three assets: self-esteem (*p* < 0.001); self-efficacy (*p* < 0.001); and problem-solving [two sub-scales (*p* = 0.002 and *p* < 0.001)].

#### 3.3.6. Lifestyle Change

96% participants reported making a lifestyle change after program participation and 93% agreed or strongly agreed that the program helped them make that lifestyle change. See Section 3.5 (Qualitative Analysis) for details of lifestyle changes made.

#### 3.3.7. Inter-Site Outcome Differences

The authors found differences in which outcome measures improved significantly between sites as detailed in Table 3. While knowledge and patient activation scores improved significantly for all sites, other measures varied in by site. For example, participants in San Jose significantly decreased consumption of sugary drinks and foods but participants in Lawrence County and Central Valley did not. Participants in Lawrence County significantly reduced consumption of fatty foods but that change was not evident in San Jose or Central Valley. Participants in Lawrence County and Central Valley reported significant improvements in psycho-social assets (self-esteem, self-worth, problem solving), while participants in San Jose did not.

### 3.4. Results of Multivariate Linear Regression

The authors compared the change in health knowledge, patient activation, health communication and understanding, health behaviors and youth psychosocial assets between three locations. In addition, the authors validated the results using Wilcoxon signed rank tests and found the same outcomes. For each pair-wise comparison (location A vs. B, location B vs. C and location A vs. C), we used the analysis of covariance (ANCOVA) with the change in outcomes of interest as dependent variable, location as independent variable of interest and grade and gender as additional independent variables adjusted for in the multiple linear regression. The authors created multivariate linear regression models to assess changes in health knowledge, patient activation, health communication and understanding, health behaviors and youth psychosocial assets. The authors calculated difference in post and pre-test scores, as the dependent variable and analyzed whether there were differences in outcomes by location when controlling for baseline (i.e., pre-test) score, grade, and gender. Other independent variables were also explored, but since they correlated with the main independent or controlling variables, those results are not shown here. For example, the authors found correlation between ethnicity and location, hence developed two different sets of multivariate linear regression models. When the authors replaced location with ethnicity, ethnicity was not a significant predictor of outcomes when controlling for baseline score, grade, and gender.

In five outcomes there was no significant difference by location when predicting outcome improvements, meaning that improvements in outcomes were not statistically different among the three locations when controlling for baseline score, grade, and gender. These outcomes were: change in health knowledge, patient activation, consumption of fruits and vegetables, ability to reduce stress and youth psycho-social asset of self-esteem (Table 4).

In six outcomes, significant differences by location were found when controlling for baseline score, grade, and gender. In health communication and understanding, in comparison with San Jose, CA and Lawrence County, MO, participants in Central Valley, CA significantly improved talking about health with family members and significantly improved understanding of what it takes to be healthy. Additionally, participants in 9^th^ and 10^th^ grade were significantly more likely to improve talking about health and understanding of health compared to older participants in 11th and 12th grades. Regarding physical activity, participants in Lawrence County, MO were significantly more likely to increase physical activity compared to participants in San Jose, CA or Central Valley, CA. Consumption of sugary foods and sugary drinks significantly decreased in San Jose, CA but not in Lawrence County, MO and Central Valley, CA. For psychosocial assets of resilience (self-efficacy and problem solving combined), participants in rural Central Valley were significantly more likely to increase youth assets of resilience compared to participants in San Jose, CA and Lawrence County, MO sites when controlling for baseline scores, grade, and gender (Table 4).

The same models were replicated using ethnicity and person coached as predictors and they did not make a significant difference in outcomes.

### 3.5. Qualitative Analysis

Participants were asked to complete open-ended questions about their experience during the SYDCP program in the post-survey. Researchers analyzed one specific open-ended question *“What specific lifestyle changes have you made* [as a result of program participation]?”

Four themes emerged from analysis of responses to the open-ended question:

Theme 1—Healthy eating: Nearly 60% participants reported changing eating habits, including increased consumption of fruits and vegetables, reduced consumption of junk food and reduction in consumption of sugary drinks, sodas and sugary foods. A participant in the Central Valley wrote:


*One of the action plans that I made as a part of this program was healthy eating, which was based off of the plate rule (1/2 low carbs veggies, 1/4 protein rich foods, and 1/4 high in carbs). I started with 2 meals a day with this method, at least 4–5 days a week at my designated eating times. I meal prepped so it was less stressful, and it worked very well!*


Theme 2—Increased physical activity: 48% participants reported exercising more, including utilizing time management skills to make time for exercise. A participant in the Central Valley wrote:


*One action plan is to exercise more with my aunt. For example, we went to walk every day for about 1 h and did zumba for 30 min after. Also, to eat healthier, we both stopped eating junk food 4 days a week and started drinking 3 water bottles every day.*


Theme 3—Improved sleep: Nearly 35% participants reported increasing hours of sleep and described adjusting sleep schedules, reducing cell phone use before bedtime, and improving quality of sleep. A participant from San Jose wrote:


*I realized how much sleep could affect my lifestyle and life span. I am starting to get more sleep and prioritizing my night routine. I tried a sleep mask and going to bed earlier. I now get around 7–8 h of sleep which is a big change from before.*


Theme 4—Stress reduction: 14% participants reported reducing stress and described incorporating breathing exercises and meditation, as well as practicing yoga. Many participants reported making numerous lifestyle changes because of program participation. A participant in Lawrence County noted:


*I start eating less snacks in between meals and started being more active by playing more with my dogs.*


A participant in Central Valley described extensive changes made in the process of working with her grandfather:


*As a part of this program, I created various action plans for my grandfather and I. For my grandfather, we worked on a walking plan to incorporate more walking and stretching into his schedule because he has to sit for the entire duration of his work on a tractor, he decided on splitting a bit of time during his lunches to stretch and take a light walk around the tractor, then once at home off of work, another walk around the ranch, he made this plan to fit into 6 out of the 7 days. We also made an action plan to incorporate daily breathing techniques that he would incorporate into the beginning of the day before breakfast and at the end of the day right before bed which was reminded with an alarm. For myself, I made an action plan to start running five times a week after a light breakfast at 6 am. I cut down my brownie/cookie intake only having 1–2 after lunch every other day. I also increased the amount of water I drank by replacing soda with water with lemon.*


Another participant in Central Valley reports making numerous lifestyle changes:


*Changes I have made are eating healthier, going to bed earlier and putting my phone down at least 10 min before going to sleep, and reducing stress by taking time to breathe. When it comes to healthier eating, I have started to not buy snack foods like chips and I have started reading food labels. With reducing stress, I have made sure to wake up 5 min earlier to take a couple deep breaths before getting ready for the day.*


Other reported lifestyle changes included reading more, procrastinating less, and getting more involved in their parents’ healthcare.

### 3.6. Additional Participant Feedback Regarding Program Benefit

When asked what participants liked about the program, 96% participants agreed or strongly agreed that they learned something new about diabetes and 92% that the program helped them connect with the family member they coached.

## 4. Discussion

This study demonstrates that remote implementation of a health promotion program for vulnerable youth is feasible and has the potential to empower at-risk youth from a variety of demographic and geographic backgrounds. After remote implementation, overall, program participants demonstrated statistically significant improvements in health knowledge, patient activation scores, health communication and understanding, health behaviors, and psycho-social assets. Additionally, 96% of participants reported making a lifestyle change to improve their health after program participation.

When comparing results between the three sites, it is notable that participants at each site demonstrated statistically significant improvements in health knowledge and patient activation scores. The improvements in patient activation scores and resultant increases in patient activation levels of study participants are particularly meaningful because higher patient activation is strongly associated with better health outcomes [40], decreased healthcare costs [45], and increased patient satisfaction with care provided [46]. PAM^®^10 scoring places individuals into one of four activation levels which correlate with patient motivation and readiness for change with patients at Level 1 disengaged and overwhelmed, lacking confidence and skills necessary to manage their own health and patients at Level 4 adopting and maintaining new behaviors and a healthy lifestyle [41]. Among study participants, dramatic increases in patient activation levels were noted. These shifts in activation levels are important because moving up a level of activation is correlated with sustained behavior change [47].

The improvements in participants’ understanding of how to improve their health speaks to the practical knowledge and skills learned during the program. Concordantly, as participants gained confidence in their ability to improve their health, they also reported increased family communication about health. Increased family communication about health has been shown to improve health beliefs and behaviors [48] which is another indication that program participation supports vulnerable youth to engage in healthy behaviors. Additionally, psycho-social assets for youth have been identified as important precursors for engaging in healthy behaviors [49,50]. The significant improvements reported in nutrition are important because of the known association between adolescent nutrition and adolescent growth and development as well as the association between nutrition and body mass index [51]. Given the dramatic rise in BMI for adolescents during the COVID-19 pandemic [11], these findings are especially meaningful as they suggest program participation could play a role in helping adolescents make healthy food choices and prevent or reduce obesity. Lastly, because adolescents experience high levels of stress that can lead to long-term physical and mental health problems [52,53], the significant improvements reported in stress reduction are noteworthy and suggest program participation could support vulnerable youth to reduce their stress levels.

Our results indicate that in most outcomes including health knowledge, patient activation, consumption of fruits and vegetables, ability to reduce stress and youth self-esteem the improvements were statistically significant among the three locations, and there was no statistical difference between locations when controlling for gender and grade. However, in some outcomes, specifically health communication and understanding, and self-efficacy and problem solving, there were some statistically significant differences between locations, and greater improvements were observed in Central Valley, CA. On the other hand, regarding consumption of sugary foods, participants in San Jose, CA were more likely to decrease consumption than participants in Central Valley, CA or Lawrence County, MO.

Overall, our study shows that the remote implementation of the SYDCP has benefit for vulnerable adolescents from a variety of settings. Although the participants were all vulnerable youth secondary to the socio-economic status of their families and communities, the three implementation sites varied by geography and demographics. Additionally, program instructors had different educational backgrounds across sites. Findings suggest that the program had benefit to participants in different ways depending on site of implementation which suggests the program can support youth from varied backgrounds and utilizing instructors with different health educational backgrounds. Understanding why significant improvements were seen for certain measures at some but not all sites requires further investigation.

Another important finding was how almost all participants reported making lifestyle changes to improve their health because of program participation. Those findings validate the significant improvements seen in the quantitative outcome measures and the qualitative data that provided examples of the specific changes made. One area in which the qualitative data did not correlate with the quantitative outcome measures was improvement in physical activity. Although many participants reported increases in physical activity in their open-ended survey responses, analysis did not yield statistically significant improvements in the quantitative outcome measures related to physical activity. One explanation for this discrepancy may be that the quantitative physical activity questions were too limited in their wording and answer choices and were not able to capture the changes that participants made.

Technology challenges were more common in rural locations (Lawrence County, MO and Central Valley, CA). This discrepancy was not surprising as there continues to be a significant gap in home broad band access between rural residents and urban residents [54]. Our participants in San Jose, CA were all able to log-in and all had devices at home, with only a few participants each week who were unable to keep their video on due to connectivity issues. Although they reported more difficulty in accessing remote technology, rural high school participants (Lawrence County, MO and Central Valley, CA) still showed benefits of participating in remote SYDCP. This success despite obstacles may be explained by the fact that SYDCP promotes regular youth-adult dyadic connections for healthy lifestyle behaviors in addition to health education. The adult connectedness may contribute to youth benefits in addition to the health knowledge gained by SYDCP participation.

Although there are numerous documented successful web-based health promotion programs for youth, in review of the literature, no studies were found that explore the efficacy of validated health promotion programs for youth adapted from in-person to synchronous remote implementation. A few studies explore the process and feasibility of adapting programs for remote implementation. For example, one recent study examined the feasibility and acceptability of adapting a sexual health program for virtual implementation for Native American teens [55]. Similar to the SYDCP experience, their program, Respecting the Circle of Life, was rapidly adapted for remote implementation by Patel et al. Despite the new model, the authors found the program implementation was feasible and was deemed acceptable by both the adolescent participants and their trusted adult guardians. Another similarity was that internet connectivity presented a challenge for some of their participants.

In another study in which a different teen sexual health program targeting Latino youth, El Camino, was adapted for a virtual setting, the authors focused on lessons learned during remote implementation [56]. Although the SYDCP team did not focus on capturing lessons learned during remote implementation for this study, many of the reported lessons learned from the El Camino virtual implementation resonate with the SYDCP team’s experience. Some of the notable lessons learned include the importance of tailoring implementation instructions to the virtual setting utilized (i.e., Zoom, Microsoft teams…etc.); having more than one facilitator which helps alleviate technical challenges; recognizing that building rapport with youth in a virtual setting requires time, energy, and creativity; and understanding youth participants may not have a private setting in which to view and participant in the program.

Another example of adapting an in-person health program for vulnerable youth during the COVID-19 pandemic is the suicide prevention program, PC CARES which targets Alaska Native youth [57]. Wells et al. detail the process of adapting PC CARES curriculum and implementation for synchronous remote implementation. While they do not report on participants’ experiences with the remote implementation, the authors highlight important considerations in the adaptation process which the SYDCP team also utilized including focusing on the needs of the community one is serving at the time; allowing for continual iterative improvements as the adapted curriculum is tested; and changing means of facilitating the program to accommodate the technological aspects of implementation.

## 5. Study Limitations

At the outset of the COVID-19 pandemic, SYDCP community partners requested an option to provide the program remotely. The SYDCP research team immediately adapted the SYDCP program for remote implementation and offered the remote program to three community partners in different settings around the United States to engage youth in health promotion despite the school cancellation that resulted from the pandemic.

Because this study arose from the unexpected and needed adaptation of the SYDCP during the pandemic, we had to work within the realities of the situation to plan the study and assess the efficacy of the program. Because the remote adaptation of the program was implemented quickly in response to community need, there was not enough time to perform a randomized controlled study or even a quasi-experimental design.

As a result, this study is limited in numerous ways. First, selection bias exists because students either elected to participate or were recruited by teachers to participate. For that reason, our results are not generalizable to all vulnerable youth. Second, because 36% of our study enrollees did not complete post-surveys, our data may be skewed towards participants who are more motivated or who benefited more from the program or who had better access to Zoom technology. Results may not be generalizable to male youth due to limited post-survey completions by male students. Third, although the study demonstrated that the program can be taught successfully by instructors with different medical backgrounds, it is likely that the instructor difference led to some variability in how the program was administered. Fourth, post participation evaluation only occurred immediately after the eight-week program and therefore could not assess duration of program benefits. Fifth, our research team did not assess program impact for the family members being coached and can thus only draw conclusions about program impact for the adolescent participants. Lastly, as explained above, this study did not include a control group which makes it difficult to determine whether our results are a direct consequence of program participation or whether other variables affected the outcome measures.

Future studies will include control groups; aim to better engage and measure program effectiveness for vulnerable male youth; and assess whether program benefit is sustained beyond program participation.

## 6. Conclusions

Programs are needed to support and empower vulnerable youth to mitigate the rise in chronic illness and disproportionate burden of disease in low-income and racial and ethnic minority populations. This study demonstrates that even in circumstances when in-person programmatic activities are not possible, remote synchronous participation in school-based health promotion and health coaching programs such as SYDCP has potential to support vulnerable youth from a variety of geographic settings and demographic backgrounds to improve health knowledge, patient activation scores and levels, health communication and understanding, health behaviors, and psycho-social assets. Program participation may also encourage adoption of lifestyle changes to improve health. Thus, it is possible that youth health programs developed for in-person implementation can be adapted successfully for remote participation and yield significant benefit for participants.

## Figures and Tables

**Figure 1 ijerph-20-01044-f001:**
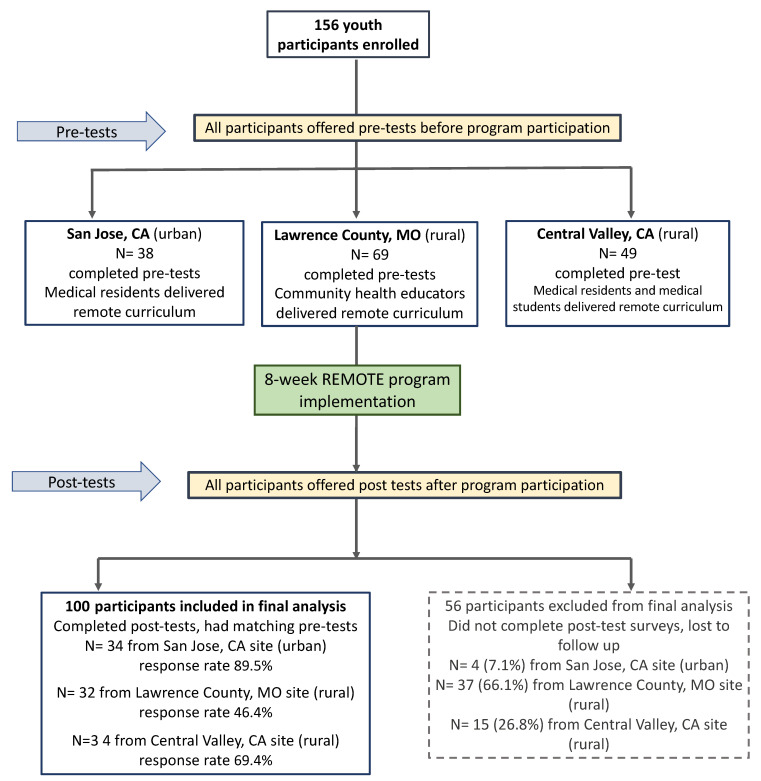
Remote implementation and evaluation of the Stanford Youth Diabetes Coaches Program (SYDPC), a health promotion and coaching skills program, in San Jose, CA; Lawrence County, MO; and Central Valley, CA, 2020–21.

**Figure 2 ijerph-20-01044-f002:**
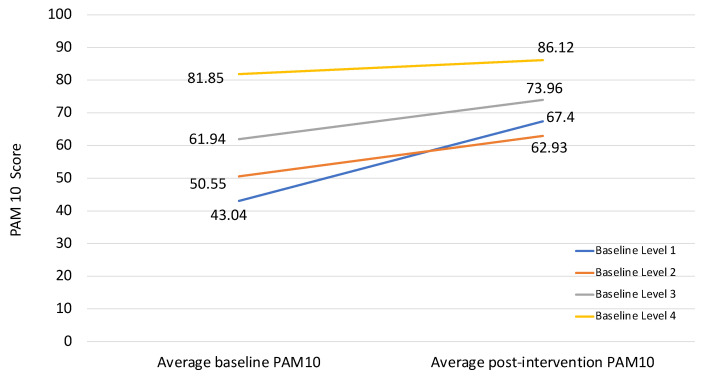
Change in PAM^®^ 10 scores from average baseline level to post-intervention level for SYDCP remote participants, (*n* = 92), San Jose, CA; Lawrence County, MO; Central Valley, CA; 2020–21.

**Table 1 ijerph-20-01044-t001:** Study Outcomes and Measures Used to Assess Outcomes for Participant Surveys Before and After Remote Intervention; San Jose, CA; Lawrence County, MO; and Central Valley, CA, 2020–21.

Outcomes	Measures Used
**Health Knowledge:** Change in health knowledge (general and diabetes-related)	8 questions adapted from Diabetes Knowledge Test by U. Michigan’s Diabetes Institute [33], the Spoken Knowledge in Low Literacy in Diabetes scale [34], and questions developed by authors [22]
**Patient Activation** Change in patient activation scores and levels	10-item Patient Activation Measure PAM ^®^ 10 from Insignia Health [35]
**Health Communication and Understanding**	3 questions developed by the authors [36]
**Health Behavior:** Change in daily physical activity Change in daily consumption of cups of fruits and vegetables Change in consumption of sugary drinks/foods and fatty foods Change in ability to manage stress	7 questions adapted from California Healthy Kids Survey Physical Health Module 2021 [37] and Stanford Mind and Body Lab [38]
**Youth Assets:** **Self-Esteem** Change in self-esteem	10-item Rosenberg Scale for self-esteem [39]
**Self-efficacy** Change in self-efficacy	4 questions adapted from California Healthy Kids’ Survey [37]
**Problem-solving** Change in problem solving ability	3 questions adapted from California Healthy Kids’ Survey [37]
**Lifestyle Change:** Specific healthy behavior change(s)	1 open-ended question developed by authors to qualitatively analyze lifestyle change(s) made after program participation

**Table 2 ijerph-20-01044-t002:** Demographic characteristics of all participants (*n* = 100) combined and by location (San Jose, CA; Lawrence County, MO; and Central Valley, CA), 2020–21.

	Combined Sample *n* = 100	Urban Site San Jose, CA *n* = 34	Rural Site Lawrence County, MO *n* = 32	Rural Site Central Valley, CA *n* = 34
Gender	*n* (%)	*n* (%)	*n* (%)	*n* (%)
Male	16 (16.0%)	3 (8.8%)	6 (18.8%)	7 (20.6%)
Female	84 (84.0%)	31 (91.2%)	26 (81.3%)	27 (79.4%)
Grade				
9th	16 (16.0%)	9 (26.5%)	3 (9.4%)	4 (11.8%)
10th	14 (14.0%)	10 (29.4%)	3 (9.4%)	1 (2.9%)
11th	45 (45.0%)	4 (11.8%)	15 (46.9%)	26 (76.5%)
12th	25 (25.0%)	11 (32.4%)	11 (34.4%)	3 (8.8%)
Age (mean age in years)	16.05 years	15.47 years	16.5 years	16.21 years
Ethnicity				
Hispanic or Latino	40 (40%)	7 (20.6%)	15 (46.9%)	19 (55.9%)
Race				
American Indian or Alaska Native	1 (1%)	0 (0%)	0 (0%)	1(2.9%)
Asian	28 (28.0%)	22 (64.7%)	1 (3.1%)	5 (14.7%)
Black or African American	3 (3.0%)	2 (5.9%)	0 (0%)	1 (2.9%)
Native Hawaiian or Pacific Islander	1 (1%)	0 (0%)	0 (0%)	1 (2.9%)
White	37 (37.0%)	1 (2.9%)	20 (62.5%)	16 (47.1%)
Two or more races	7 (7.0%)	4 (11.8%)	1 (3.1%)	2 (5.9%)
Declined to respond	23 (23.0%)	5 (14.7%)	10 (31.3%)	8 (23.5%)
Other Sample Characteristics				
Live within 15 min to place where shop for food	77 (77%)	25 (73.5%)	29 (90.6%)	23 (67.6%)
Access to fresh fruits and vegetables most times	93 (93%)	31 (91.2%)	31 (96.9%)	31 (91.2%)
Migrant work not main source of family income	87 (87%)	31 (91.2%)	28 (87.5%)	28 (82.4%)
Person Coached *				
Parent	45 (45 %)	12 (35.3%)	16 (50%)	17 (50%)
Other family member	31 (31%)	11(32.4%)	9 (28.1%)	11 (32.4%)
Friend or other	21 (21.0%)	11 (32.4%)	6 (18.8%)	4 (11.8%)
Person coached had diabetes	26 (26.0%)	7 (20.6%)	10 (31.3%)	9 (26.5%)

* Data missing for 3 participants who did not respond to this question.

**Table 3 ijerph-20-01044-t003:** Pre-Post Mean Differences in Outcome Measures of All Participants Combined (*n* = 100), and by site (San Jose, CA; Lawrence County, MO; and Central Valley, CA), 2020–21.

	All Combined *n* = 100	San Jose, CA (Urban) *n* = 34	Rural Site Lawrence County, MO *n* = 32	Rural Site Central Valley, CA *n* = 34
**Evaluation Measures/Outcomes**	Mean difference (SD)	Mean difference (SD)	Mean difference (SD)	Mean difference (SD)
^1^ Health Knowledge	**3.55 (2.08) ****	**3.676 (1.6) ****	**3.75 (2.1) ****	**3.235 (2.4) ****
**Patient Activation Measure (*n* = 92)**				
^2^ PAM 10^®^ mean scores	**11.66 (15.05) ****	**9.22 (15.7) ****	**10.72 (15.5) ****	**14.76 (13.8) ****
**Health communication and understanding**				
^3^ Talking about health at home	**0.230 (0.89) ***	0.059 (0.92)	0.219 (0.75)	**0.412 (0.98) ***
^4^ Understanding health improvement	**0.290 (0.91) ****	0.176 (0.99)	0.281 (1.05)	**0.412 (0.65) ****
**Health Behaviors**				
^5^ Physical Activity	0.270 (1.99)	−0.088 (1.5)	0.656 (2.4)	0.265 (1.9)
^6^ Fruit and vegetable consumption	**0.290 (0.98) ****	0.147 (0.96)	**0.438 (1.16) ***	**0.294 (0.83) ***
^7^ Consumption of sugary drinks	−0.18 (1.03)	**−0.353 (0.98) ***	−0.062 (0.94)	−0.118 (1.15)
^8^ Consumption of sugary foods	−0.17 (1.3)	**−0.382 (0.98) ***	−0.094 (1.2)	−0.029 (1.22)
^9^ Consumption of fatty foods	−0.1 (1.0)	−0.147 (.82)	**−0.406 (1.0) ***	0.235 (1.1)
^10^ Ability to reduce stress	**0.360 (1.18) ****	0.176 (1.06)	0.313 (1.3)	**0.588 (1.2) ***
**Youth Assets**				
^11^ Self-esteem	**1.2 (3.47) ****	0.265 (3.5)	**1.438 (3.09) ***	**1.912 (3.6) ****
^12^ Self-efficacy	**0.810 (2.03) ****	0.353 (2.3)	**0.844 (1.7) ***	**1.235 (1.94) ****
^13^ Problem solving (2 questions)	**0.47 (1.49) ****	0.088 (1.5)	**0.50 (1.3) ***	**0.824 (1.5) ****
^14^ Problem solving (1 question)	**0.290 (0.74) ****	0.235 (0.69)	0.188 (0.78)	**0.441 (0.74) ****

* *p* value < 0.05, ** *p* value < 0.005 using paired *t*-tests SPSS version 27. Score range ^1^ 0–8, ^2^ 0–100, ^3^ 1–5, ^4^ 1–5, ^5^ 0–7, ^6^ 0–5, ^7^ 0–5, ^8^ 0–5, ^9^ 0–5, ^10^ 1–5, ^11^ 10–40, ^12^ 4–16, ^13^ 2–8, ^14^ 1–4.

**Table 4 ijerph-20-01044-t004:** Linear regression models for select outcomes in San Jose, CA; Lawrence County, MO; and Central Valley, CA; 2020–21. (*n* = 100).

Outcome Category	Predictor Variables	Coefficient (SE)	Confidence Intervals (Lower, Upper)
**Health Knowledge**	Pre-test score	**−0.756 (0.112) ****	−0.977, −0.534
	Grade (ref = 9th and 10th grade)	0.341 (0.422)	−0.496, 1.179
	Gender (ref = male)	0.149 (0.482)	−0.809, 1.106
	Location (ref = urban)		
	Rural MO	−0.679 (0.475)	−1.622, 0.264
	Rural Central Valley	−0.320 (0.462)	−1.237, 0.598
**Patient Activation Measure *(n* = 92)**			
PAM^®^10 (*n* = 92)	PAM^®^10 pre-test score	**0.782 (0.142) ****	0.499, 1.064
	Grade (ref = 9th and 10th grade)	0.111 (3.835)	−7.513, 7.736
	Gender (ref = male)	4.67 (4.446)	−4.17, 13.508
	Location (ref = urban)		
	Rural MO	1.974 (4.24)	−6.46, 10.407
	Rural Central Valley	6.563 (4.12)	−1.623, 14.75
**Health communication and understanding**			
Talking about health at home	Pre-test score	**−0.703 (0.080) ****	−0.863, −0.543
	Grade (ref = 9th and 10th grade)	**−0.340 (0.158) ***	−0.654, −0.026
	Gender (ref = male)	0.026 (0.185)	−0.340, 0.393
	Location (ref = urban)		
	Rural MO	0.170 (0.175)	−0.178.517,
	Rural Central Valley	**0.641 (0.175) ****	0.293, 0.988
Understanding of health	Pre-test score	**−0.930 (0.109) ****	−1.146, −0.714
	Grade (ref = 9th and 10th grade)	**−0.376 (0.164) ***	−0.701,−0.051
	Gender (ref = male)	−0.008 (0.190)	−0.385, 0.369
	Location (ref = urban)		
	Rural MO	0.100 (0.181)	−0.260, 0.460
	Rural Central Valley	**0.471 (0.180) ***	0.113, 0.830
**Health Behaviors**			
Physical Activity	Pre-test score	**−0.685 (0.082) ****	−0.847, −0.522
	Grade (ref = 9th and 10th grade)	−0.001 (0.366)	−0.728, 0.726
	Gender (ref = male)	−0.070 (0.428)	−0.919, 0.779
	Location (ref = urban)		
	Rural MO	**0.865 (0.403) ***	0.064, 1.666
	Rural Central Valley	0.587 (0.405)	−0.218, 1.391
Consumption of fruits and vegetables	Pre-test score	**−0.612 (0.093) ****	−0.796, −0.427
	Grade (ref = 9th and 10th grade)	0.049 (.199)	−0.347, 0.445
	Gender (ref = male)	0.083 (.230)	−0.373, 0.540
	Location (ref = urban)		
	Rural MO	−0.131 (0.228)	−0.584, 0.322
	Rural Central Valley	−0.097 (0.223)	−0.540, 0.346
Consumption of sugary drinks	Pre-test score	**−0.588 (.073) ****	−0.733, −0.442
	Grade (ref = 9th and 10th grade)	−0.069 (0.192)	−0.449, 0.311
	Gender (ref = male)	−0.235 (0.220)	−0.673, 0.202
	Location (ref = urban)		
	Rural MO	0.550 (0.211)	0.130, 0.970
	Rural Central Valley	0.564 (0.213)	0.141, 0.988
Consumption of sugary foods	Pre-test score	**−0.676 (0.082) ****	−0.839, −0.512
	Grade (ref = 9th and 10th grade)	0.022 (0.209)	−0.394, 0.438
	Gender (ref = male)	0.371 (0.242)	−0.109, 0.851
	Location (ref = urban)		
	Rural MO	0.470 (0.231)	0.012, 0.929
	Rural Central Valley	0.626 (0.233)	0.164, 1.088
Stress Reduction	Pre-test score	**0.722 (0.127) ****	−0.975,−0.469
	Grade (ref = 9th and 10th grade)	0.220 (0.244)	−0.264, 0.704
	Gender (ref = male)	0.215 (0.287)	−0.354, 0.784
	Location (ref = urban)		
	Rural MO	−0.455 (0.284)	−1.019, 0.108
	Rural Central Valley	−0.015 (0.276)	−0.563, 0.534
**Youth Assets**			
Psychosocial Assets			
Youth Resilience (Combined)	Pre-test score	**−0.461 (0.090) ****	−0.639, −0.282
(Self-efficacy and problem solving)	Grade (ref = 9th and 10th grade)	−0.036 (0.747)	−1.519, 1.448
	Gender (ref = male)	0.319 (0.840)	−1.350, 1.988
	Location (ref = urban)		
	Rural MO	0.496 (0.803)	−1.097, 2.090
	Rural Central Valley	**1.889 (0.808) ***	0.284, 3.494
Self-Esteem	Pre-test score	**−0.250 (0.073) ****	−0.395, −0.104
	Grade (ref = 9th and 10th grade)	0.539 (0.785)	−1.020, 2.099
	Gender (ref = male)	0.629 (0.903)	−1.163, 2.421
	Location (ref = urban)		
	Rural MO	0.970 (0.862)	−0.742, 2.682
	Rural Central Valley	1.440 (0.865)	−0.277, 3.157

* *p* < 0.05, ** *p* < 0.005.

## Data Availability

All relevant data are within the paper. We respect the value in making data publicly available. Unfortunately, we cannot share individual data points due to IRB compliance because we agreed that we will not share our participant files as the data comes from a vulnerable population (ethnic minority youth) and if shared publicly, confidentiality could be breached. Questions in our survey include sensitive information including self-reported feelings of worth, grades, living situation, and descriptions of interpersonal relationships with family members. Especially given that to describe the implementation of the dissemination of the SYDCP in the paper, we reveal locations, it would be easy to determine particular schools where the program has been implemented–and accordingly identify individual participants. Additionally, the fact that we report grade, race/ethnicity, living situation and language used at home makes our participants particularly vulnerable to breaches of confidentiality if individual level data is made public. Due to these ethical obligations to our participants and to abide by the conditions of our IRB status, we are not able to publicly publish the data. If an individual would like to examine our data, s/he would need to first complete all Stanford IRB clearance requirements regarding ethical compliance and confidentiality. After that, data could be made available by making a request to our research team data analysis specialist Ashini Srivastava ashini@stanford.edu.

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
