# Peer review of "Remote Implementation of a School-Based Health Promotion and Health Coaching Program in Low-Income Urban and Rural Sites: Program Impact during the COVID-19 Pandemic"

_ijerph, 2023, doi:10.3390/ijerph20021044_

Round 1

Reviewer 1 Report (Previous Reviewer 3)

I appreciate your thoughtful responses to all comments.

I believe that the author’s addition of alternative indicators of socioeconomic status by location has helped to further clarify the key factors in this study. Please consider the following minor points.

1. I believe that the text in lines 81-85 and 94-96 of the Introduction should be moved to the Discussion (limitation). I also believe that lines 88 to 93 and Table 1 would be more appropriately stated in the Methods.

2. The sample size calculation has been added, is this the result of the entire number of participants or in each group? In any case, the required number of participants is not available for several outcome measurements. I assume that the sample size was calculated later by reviewer comments. If the calculation was made before the study started, please provide detailed reasons for setting the target of 100-150 participants, despite the insufficient number. If the calculation was made after the study started, please base this manuscript on that fact.

Author Response

Thank you for your comments. Please find responses below:

Comment 1: I believe that the text in lines 81-85 and 94-96 of the Introduction should be moved to the Discussion (limitation). I also believe that lines 88 to 93 and Table 1 would be more appropriately stated in the Methods.

We have moved lines 81-85 and 94-96 to the limitations section of the discussion. 

We have moved lines 88-93 and Table 1 to the Methods section (2.6). After moving these lines to the methods section, we added a brief explanation of outcome measures (“key health related outcomes as described in the methods section”) to the introduction and removed duplicate language describing the outcome measures in section 2.6.

Comment 2: The sample size calculation has been added, is this the result of the entire number of participants or in each group? In any case, the required number of participants is not available for several outcome measurements. I assume that the sample size was calculated later by reviewer comments. If the calculation was made before the study started, please provide detailed reasons for setting the target of 100-150 participants, despite the insufficient number. If the calculation was made after the study started, please base this manuscript on that fact.

Thank you for helping us clarify this question. First, those sample size estimations were calculated using results from previous groups to which we provided the program a few years ago. We have clarified this in the text by adding the following statement: “Sample size estimations were conducted using results from previous groups to which we provided the program a few years ago.”
In those estimations we found that a larger sample size would be needed to detect significant pre-post differences in some of the outcomes measures, yet a) we had made improvements to our curriculum since those estimations were calculated, and b) we had evidence from previous group interventions of the fact that groups that score lower at the start of the program tend to gain more and report greater pre-post improvements than students who already have higher scores to start with. In the end, our results demonstrate that even with smaller sample size in some outcomes the program still proved to make a statistically significant improvement based on the results of this study groups. Consequently, the sample calculations for future research using these measures will be lower than the ones we calculated based on previous pilot research.

Second, we state in the manuscript that “the overall aim was to enroll 100-150 students,” but upon reflection given your question, that statement does not accurately reflect the reality of this study, so we removed that statement.

As we explained in the manuscript, “As this study arose from the unexpected and needed adaptation of the SYDCP during the pandemic, we had to work within the realities of the situation to plan the study and assess the efficacy of the program. Because the remote adaptation of the program was implemented quickly in response to community need, there was not enough time to perform a randomized controlled study or even a quasi-experimental design.” For the reasons stated, we were also not able to predetermine our sample size for the study. Rather, we accepted interested high school student participants at each site in the midst of pandemic chaos and asked them to complete the pre and post surveys.

This manuscript is a resubmission of an earlier submission. The following is a list of the peer review reports and author responses from that submission.

Round 1

Reviewer 1 Report

Dear authors,

Even if your paper is good, the lack of formatting influences a lot the quality of the paper.

lines 48-71 are having different fonts

please do not use bullets in tables

lines 281-287 the text needs a supporting table

you used a lot of subtitles, these should be numbered according to the journals' requests

references are not formatted etc.

The results must be rewritten more clearly, the high number of subtitles is very disturbing and refer to different analysis used instead of the main issues approached in the study

please use the third person since it should be an impersonal paper.

the conclusion section is very poor, please add more information

Please revise these aspects

Reviewer 2 Report

Firstly, I want to congratulate the authors for conducting such an interesting study. The purpose of this study was to examine the impact of implementing a school-based health promotion program adapted for synchronous remote learning for adolescent participants from rural and urban underserved communities. The study is novel; however, it poses major problems.
Introduction:
- The introduction on the importance of the study is lacking, and how it is linked to COVID19 and the importance of this program.
- There are many “cut and paste” in the manuscript. Please correct it. 
- After reading the introduction, I could not get any information on the SYDCP program. 
Please justify the importance of this program during COVID-19, since the program was implemented 12 years ago.

Methods:
- Can you please define urban and rural? Such as population? Each country has slight differences. 
- Please explain why 56 participants did not complete the post-test survey? The reasons for summer break and no follow-up are not good excuses. 
- Seemed like it is a purposive design rather than random sampling in the schools for recruitment? Please elaborate. 
- Please provide sample size determination.
- How to recruit participants?
- Please provide an ethical approval code, and did the researchers receive informed consent? Another issue I observed is that the 21910 Stanford IRB title is so different from this study; please justify? Is it a salami-slicing study?
- There are so many questionnaires used; please mention the validity and reliability of the questionnaires used. 
- Analysis – it is wrong, I suggest using mixed factorial ANOVA as its t-test will cause a type-1 error. 
- For the qualitative interview, who conducted the analysis? How was it done? How was it grouped into four themes? Missing information again. 

Results: 
- The figure 2 illustration is wrong. What is pre-test levels 1, 2, 3 and 4? 

Discussion: 
Discussion is repeating on the results rather comparing with other previous studies. 
In limitation, the researcher suggested many limitations; however, I don’t understand why it was not considered in the first place? 

Conclusion:
I do not agree with the conclusion. Please revise. 

Reviewer 3 Report

This is a well-written study with important implications, which examines whether remote implementation has positive impacts on health for youth in low-income and/or racial and ethnic minority populations. However, there is a discrepancy between the objectives and the methods, which seems insufficient to conduct the intended assessment. I had the following comments requiring clarification and suggestions to improve the manuscript.

Major comments

  1. As stated by the authors in the limitations, the lack of a control group is a major limitation of this study. I recommend that any interventional study (even if it is not a randomized controlled trial) be reported following CONSORT and the CONSORT NPT Extension.
  2. What is the main outcome of this study? By presenting the main indicators of this study that SYDCP targets, the program outline and its effectiveness will be clearer.
  3. Despite the target population of this study being economically disadvantaged youth, the information on the socioeconomic status of the participants (who were analyzed) is vague. As the author mentioned in the limitations section, health-conscious high school students in San Jose and Lawrence participated in the study, 93% of the participants had access to fresh fruits and vegetables at most times, and many of the students who dropped out were families of migrant workers, which suggests that it is likely that the participants in this study were not economically disadvantaged. Therefore, the assumption of "disadvantaged" may not be valid. The authors need to be more cautious in their interpretation of the results and conclusions.
  4. I have the impression that using location as an explanatory variable in multivariate analysis is misaligned with the purpose. To examine whether there is an effect of remote implementation for economically advantaged youth, it is necessary to compare the difference in effect by socioeconomic status. It would be interesting to analyze the results by the socioeconomic status of the participants.
  5. Qualitative analysis has also been reported, but I believe that the significance of using qualitative data is to know the meaning, not the frequency or trend. Therefore, it is necessary to also evaluate before the intervention (baseline) and examine the qualitative changes before and after the intervention. I also believe that reporting the 4% negative (not positive) writing will help prevent dropouts and lead to the development of new intervention programs that leave no one behind when conducting the future study.

Minor comments

  1. (Abstract): I believe that be easier to understand the effect if there are concrete figures of the main quantitative results in the abstract.
  2. (Lines 63-65 and 424-426): The fonts are different.
  3. (Lines 101-109 and Figure 1): What is the overall number of students authors recruited? Please revise the text and the flow diagram to show the consent rate (participation rate) for each site.
  4. Tables 2 and 3: I believe the readability would be enhanced if the Evaluation measures/Outcomes were left-justified, and the numbers were right-justified with aligned decimals (SD cells should be separated).